# How large are temporal representativeness errors in paleoclimatology?

### Daniel E. Amrhein<sup>1</sup>

5

<sup>1</sup>University of Washington School of Oceanography and Department of Atmospheric Sciences **Correspondence:** Dan Amrhein (amrhein@uw.edu)

Abstract. Ongoing work in paleoclimate reconstruction prioritizes understanding the origins and magnitudes of errors that arise when comparing models and data. One class of such errors arises from assumptions of proxy temporal representativeness (TR), i.e. how accurately proxy measurements represent climate variables at particular times and time intervals. Here we consider effects arising when 1) the time interval over which data average and the climate interval of interest have different durations, 2) those intervals are offset from one another in time (including when those offsets are unknown due to chronological uncertainty), and 3) the paleoclimate archive has been smoothed in time prior to sampling. Because all proxy measurements are time averages of one sort or another, and it is challenging to tailor proxy measurements to precise time intervals, such errors are expected

to be common in model-data and data-data comparisons, but how large and prevalent they are is unclear. This work provides a first-order quantification of temporal representativity errors and to studies the interacting effects of sampling procedures,
archive smoothing, chronological offsets and errors (e.g. arising from radiocarbon dating), and the spectral character of the climate process being sampled.

Experiments with paleoclimate observations and synthetic time series reveals that TR errors can be large relative to paleoclimate signals of interest, particularly when the time duration sampled by observations is very large or small relative to the target time duration. Archive smoothing can reduce sampling errors by acting as an anti-aliasing filter, but destroys high-frequency

- 15 climate information. The contribution from stochastic chronological errors is qualitatively similar to that when an observation has a fixed time offset from the target. An extension of the approach to paleoclimate time series, which are sequences of time-average values, shows that measurement intervals shorter than the spacing between samples lead to errors, absent compensating effects from archive smoothing. Nonstationarity in time series, sampling procedures, and archive smoothing can lead to changes in TR errors in time. Including these sources of uncertainty will improve accuracy in model-data com-
- 20 parisons and data comparisons and syntheses. Moreover, because sampling procedures emerge as important parameters in uncertainty quantification, reporting salient information about how records are processed and assessments of archive smoothing and chronological uncertainties alongside published data is important to be able to use records to their maximum potential in paleoclimate reconstruction and data assimilation.

#### 1 Introduction

Paleoclimate records provide important information about the variability, extremes, and sensitivity of Earth's climate to greenhouse gases on time scales longer than the instrumental period. As the number of published paleoclimate records has grown and the sophistication of numerical model representations of past climates has improved, it has become increasingly important

5 to understand the uncertainty with which paleoclimate observations represent climate variables so that they can be compared to one another and to model output. Additionally, quantifying uncertainty is important for ongoing efforts to assimilate paleoclimate data with numerical climate models (e.g., Hakim et al., 2016; Amrhein et al., 2018).

Paleoclimate records can have errors arising from many different sources: biological effects (e.g., Elderfield et al., 2002; Adkins et al., 2003), aliasing onto seasonal cycles (Wunsch, 2000; Fairchild et al., 2006; Dolman and Laepple, 2018), spatial

- representativeness (Van Sebille et al., 2015), proxy-climate calibrations (e.g., Tierney and Tingley, 2014), and instrument 10 errors, to name a few. This paper focuses on errors from temporal representativeness (TR), which we define as the degree to which a measurement averaging over one time interval can be used to represent a second, target time interval. For instance, in a data assimilation procedure that fits a model to observations at every year, it is important to know the uncertainty associated with relating a decadal-average proxy observation to an annual-average target interval. Furthermore, computing a mean is
- often the implicit goal of binning procedures that combine observations from within a target time period such as a marine 15 isotope stage, and we expect those observations to have errors that vary with their averaging duration and offsets from the target. Importantly, the term "error" is not meant to connote a procedural error on behalf of a collector or user of observations: Given the sparsity of data and the nature of geophysical time series, there is often a good rationale to use one time period to approximate another that is adjacent or has a different duration. Our goal is to understand the uncertainty arising in such a
- 20 representation in a general framework.

Much of the previous study of errors arising from sampling in time has focused on aliasing, whereby variability at one frequency in a climate process appears at a different frequency in discrete samples of that process. Pisias and Mix (1988) described consequences of aliasing in the study of deterministic peaks in climate spectra due to Milankovich forcing. Wunsch and Gunn (2003) described criteria for choosing sample spacing so as not to alias low-frequency variability in sediment cores,

- 25 and Wunsch (2000) demonstrated how aliasing can lead to spurious spectral peaks in ice core records. Beer et al. (2012) and von Albedyll et al. (2017) describe how running means can reduce aliasing of solar cycle variability in ice core records. In paleoclimate, measurements are often unevenly spaced in time due to changes in archive deposition rates; Jones (1972) showed that aliasing is present and even exacerbated in unevenly-sampled records relative to regularly sampled ones. Anderson (2001) and McGee et al. (2013) describe how bioturbation and other diagenetic processes smooth records in time and may reduce 30
- aliasing errors.

A second area of previous focus stems from chronological uncertainties, whereby times assigned to measurements may be biased or uncertain. In some cases, such as for radiocarbon dating, estimates of these uncertainties are available from Bayesian approaches that incorporate sampling procedures (Buck, 2004; Buck and Millard, 2004; Bronk Ramsey, 2009); practices for incorporating this information into model-data or data-data comparisons vary, and developing tools for analyzing chronological uncertainty is an active area of research. Huybers and Wunsch (2004) include the effect of uncertainties in tie points in order to align multiple records of Pleistocene oxygen isotopes, and Haam and Huybers (2010) developed tools for estimating the statistics of time-uncertain series. The effect of time uncertainty on estimates of signal spectra is modest in some cases (Rhines and Huybers, 2011), in part because time uncertainty acts to smooth high-frequency variability when computed as an expectation over a record (Moore and Thomson, 1991).

5

This paper synthesizes effects contributing to TR errors in an analytical model and explores their amplitudes and dependence on signal spectra and sampling time scales. Extending results from time-mean measurements to time series demonstrates how sampling practices can lead to aliasing errors when records are not sampled densely, e.g. when an ocean sediment core is not sampled continuously along its accumulation axis. While we do not claim that TR error is the most important source of

10

15

uncertainty in paleoclimate records, it does appear to be large enough to affect results in some cases. Moreover, this work is a step towards reducing the number of "unknown unknowns" in paleoclimate reconstruction.

#### 2 Origins of temporal representativeness error

Our focus is first on errors arising when a mean value computed over one time period is used to represent another time period - for instance, when a time average over 20-19 kya (thousand years ago) is used to represent an average over 23-19 kya, the nominal timing of the Last Glacial Maximum (Clark et al., 2012). We define the TR error  $\theta$  as the difference between x and y

$$\theta = x - y. \tag{1}$$

where y is affected by one or more type of TR error. As illustrated using a synthetic time series in Figure 1, our focus is on TR errors arising when:

- The time interval over which an observation averages in time  $(\tau_v)$  has a different length from that of the targeted time interval ( $\tau_{\rm r}$ ; Figure 1a).
- 20 - The time interval  $\tau_v$  is offset in time from  $\tau_v$  by a time  $\Delta$  (Figure 1b). These offsets can be either known or, in the presence of chronological uncertainty of observations, stochastic and unknown.
  - The paleoclimate archive was smoothed prior to sampling, whether by by bioturbation, diagenesis, residence times in karst systems upstream of speleothems (Fairchild et al., 2006), or other effects. In order to perform a first-order exploration of smoothing effects, we represent archive smoothing moving average over a time scale  $\tau_a$ . Figure 1c illustrates how smoothing introduces errors for the case where  $\tau_a = 2\tau_x$ .
- 25

Visual inspection of Figure 1 yields some intuitive expectations. As the observational time interval  $\tau_{\rm v}$  grows small relative to  $\tau_{\rm r}$ , one expects TR errors to grow as the observation "feels" more of the variability at high frequencies. TR errors could also be expected to grow as a measurement is increasingly offset from the target in time. But interactions between different types of

errors complicate the picture: for instance, in some cases a measurement interval that is short relative to  $\tau_{\rm r}$  might have smaller

**Figure 1.** Several factors can contribute to temporal representativeness errors, defined here as the difference  $\theta$  between a true time-average paleoclimate quantity *x* and a measurement *y* that averages over a different time interval. These effects are illustrated using a synthetic autoregressive time series. In each panel, the true quantity *x* is the same. Panel (a) shows the difference when the *y* averages over a time duration  $\tau_y$  that is 5 times shorter than the averaging interval  $\tau_x$  of the target value. Panel (b) shows the error when the observed and target averaging intervals are the same, but the observation is centered on a different value in time. Additional uncertainties, not shown here but discussed in the text, arise if the time offset is a stochastic random variable, as can occur e.g. with chronological uncertainties from radiocarbon dating. Panel (c) illustrates effects when the observation spans the correct time interval, but when the paleoclimate archive being sampled stores a smoothed version of the true signal; here that smoothing has a timescale of  $\tau_a = 2\tau_x$ . These errors are merely examples and are not meant to argue, e.g., that offset errors are always greater than errors from different averaging periods.

error if it is also offset in time, or if it samples an archive that stores a smoothed version of the climate signal. Subsequent sections examine interactions between various TR error sources.

This list is not exhaustive and neglects, for instance, effects from small numbers of foraminifera in sediment core records and other errors that are inherited from the construction of r(t) from proxy observations. To isolate TR errors we assume that observations directly sample the true climate process, r(t). This approach is intended to be complementary to proxy system models (PSMs; e.g., Evans et al. (2013)) that relate proxy quantities to climate variables ("forward operators" in the language of data assimilation). The procedures described may be used to estimate TR uncertainty when PSMs do not; when they do, the model can provide a theoretical grounding for understanding those results. Variances from multiple error sources can be added together under the approximations that they are independent and Gaussian. When these assumptions fail, more holistic forward

10 modeling of errors in PSMs may be necessary.

30

#### **3** Estimating temporal representativeness error

Because in paleoclimatology we do not have complete knowledge of the underlying climate signal r(t) (it is what we are trying to sample), it is impossible to infer what the TR error is for each measurement as done in the synthetic example (Figure 1). Instead, our aim is to determine typical values for errors, which are important for data assimilation and for comparing models

15 and observations and observations to one another. We will characterize TR error  $\theta$  by estimating its variance,  $\langle (\theta - \langle \theta \rangle)^2 \rangle$ , where angle brackets denote statistical expectation. To do this, we approximate r(t) as being weakly statistically stationary, meaning that its mean and variance do not change in time; caveats surrounding this assumption are addressed later in the paper. Under the weak stationarity assumption, the mean error  $\langle \theta \rangle$  is zero, and we take the expectation by evaluating  $\theta^2$  at all the times in r(t) to compute the variance,

$$20 \quad \left\langle \theta^2 \right\rangle = \frac{1}{\tau_0} \int_{t_0}^{t_f} (x - y)^2 dt, \tag{2}$$

where  $t_0$  and  $t_f$  are the initial and final times in r(t), and  $\tau_0 = t_f - t_0$ . Intuitively, we are estimating the error in representing x by y (*at a single time*) as the time-mean squared difference of running means of r(t) (over all times). In practice, though we do not know r(t), knowledge of its statistics is adequate to estimate  $\langle \theta^2 \rangle$ .

Representing TR error in the frequency domain (Appendix A) emerges as an intuitive way to describe errors that also provides closed-form expressions that can be readily integrated to explore the effects of different sampling and time series parameters. A basic result (see Equation A13) is that in the frequency domain, TR errors are represented as

$$\left\langle \theta^{2} \right\rangle = \frac{1}{\tau_{0}} \int_{0}^{\infty} H\left( \mathbf{v}, \tau_{x}, \tau_{y}, \tau_{a}, \Delta \right) \left| \hat{r}\left( \mathbf{v} \right) \right|^{2} d\mathbf{v}.$$
(3)

where v denotes frequency, H is a so-called transfer function, and  $|\hat{r}(v)|^2$  is the power spectral density of the true signal r(t). In effect, the error variance is a weighted sum of the power at different frequencies in r(t), where the weights in frequency space (given by H) depend on how the paleoclimate record has been sampled and smoothed. This behavior is typical of aliasing,

where variance in the signal at one frequency appears erroneously in a measurement at a different frequency (in this case, at the zero frequency, which is the time mean).