# Peer review of "How large are temporal representativeness errors in paleoclimatology?"

_Climate of the Past, 2019_

## Referee Comment (RC1) · Anonymous Referee #1 · 23 Apr 2019

This study addresses an important issue, which often is forgotten in the paleodata comparisons. Paleoclimatic measurements are usually dated and mostly dating uncertainties are communicated, too. However, depending on the archive and sample methods, measurements are often time averages, integrated over a specific period. This averaging period needs to be taken into account if these measurements are compared to simulations or other observations with different averaging periods. This study makes an important contribution to quantifying this error source.

The topic is highly relevant for Climate of the Past, ideas are novel and substantial conclusion reached. I found the figures, which introduce the basics concepts, very clear and helpful illustration. However, I assume the level of text and presented mathematical background goes beyond the average reader of this journal. Therefore, I suggest

major revision, which should mainly simplify the entire text, to make it clearer and more accessible to a broad audience.

Abstract:

- Line 11: Many expressions have not been introduced, yet and are therefore not clear to the reader. What is the "target interval". Better avoid abbreviations like "tau" in the abstract.

- Line 13: What is meant by "archive smoothing" and "anti-aliasing"?

Introduction:

- Page 2, lines 3ff.: I would suggest bullet points for the various error sources

Fig. 1: Great, this makes the problem easily accessible.

Table 1: This looks more like the variable list of the 500-page book than for a CP article. It may help to have reduced (basic) version of the mathematical background in the main part of the paper and the derivations in a supplement.

2.1

- Page 6, line 1: "boxcar" needs to be explained

- Page 6, line 6: over over

Fig. 3: Labeling too small

References: Latex code remained in the pdf version.

---

## Referee Comment (RC2) · Anonymous Referee #2 · 30 Apr 2019

The paper by Armhein addresses an important topic in palaeoclimatology, namely that of time representativeness of proxy data and the subsequent use of that data to compare between models and other data sets. I found the paper very hard to read and very technical, especially for COTP, though I would consider myself at the more technical end of the audience for this type of paper. I wonder if this work might be served better by publishing the mathematics in a more theoretical journal and subsequently interpretation and case studies in COTP. I found the figures well-presented but hard to interpret, and I found the table of notation (Table 1) particularly unhelpful; it looks like it was put together in 5 minutes.

The key concept seems to revolve around the idea that there is a target measurement x which is desired to be averaged over temporal period tau_x, but the scientist only

has access to an observation y which is averaged over an interval of tau_y. The origin of both the target and the observation is the 'true' time series r(t). The paper's goal seems to be to estimate the error in using y to estimate x. I found this slightly strange, for two reasons. The first is that x and y are assumed to be centred at different time points. It's not clear to me why the centring needs to be (that) different. The second is that, if I were doing this, I would try to estimate r(t) as well as I possibly could, and then create smooths from this to allow me to compare with other time series. Perhaps this isn't possible, but I couldn't see why.

I wonder if some of these problems might be solved by having a maximally simple running example at the start that makes clear the novelties of the approach using the bare minimum of maths/notation. That seems to be what has been attempted in Figure 1, though this just raises further questions. The light grey line appears to be r(t) (this isn't mentioned?) which is unobserved. Labelling the vertical axis as r(t) is slightly confusing since y and x are not values from r(t) but averages from it. We would like to get x from y where y is centred on t minus Delta over period tau_y and x is centred on t over period tau_x. The key quantity theta is defined as the different between x and y, and most of the maths proceeds with the assumption that r(t) is weakly stationary. At this point I find myself a bit stuck as to the appropriateness of how this works. I don't think I've ever seen a situation similar to this. Aside from the stationarity assumption (which might hold locally?) in all the cases I've worked we have multiple y observations covering the spanned period. (I suppose it might be more realistic if tau_x < tau_y as often we're interested in estimating a climate value at a more precise time point than the observations).

I'm further confused by the lower panel of Fig 1 which I initially assumed was the distribution of the error described by tau_x (they nearly match in magnitude) but is actually a completely different measurement uncertainty. I don't think that panel is helpful here. However, the legend points out that the uncertainty in theta is characterised by: sampling procedures, variability of r, archive smoothing, and chronological uncertainty. I

totally agree with these fantastically important points. It's a strangely important sentence to appear in a figure caption. Having the lower panel display just one of these is confusing.

In Section 2 the paper dives into a lot of technical detail which I followed mathematically but quickly lost all the interpretation. I was hoping to pick things up again in Figure 2 which shows the frequency representation of the boxcar function at the top and the power transfer function at the bottom for assumed values of Delta, tau_x, and tau_y. This seems to show which frequencies are contributing most to the TR error. However I've read the bullet points in Section 2.3 about 5 times now and I can't follow them. They talk about certain summary statistics which aren't shown in the Figure (226 and 1325 years?).

Figure 3 rescues things a little bit by being a bit clearer but even then it doesn't introduce the 4 panels or the colours so I can only make good guesses as to what some these plots are showing. The top two panels are particularly helpful. Unfortunately without any further interpretation of Figures 2 and 3 I was completely lost beyond Section 3.1, which is a big shame. I really wanted to follow this.

Some minor points: * I got very confused about the role of tau_a. I thought the archive smoothing was represented by tau_y as that is what we are observing? * Section 2 I thought was mis-named. It's not really a statistical model in a generative sense. It's more a collection of useful summary statistics that can indicate problems in temporal representativeness. * The glossary in Table 1 needs at least a sentence of interpretation for the more complicated quantities. Saying e.g. H(v) represents the power transfer function is fine, but what would be more useful is to say that high values of this for frequencies v indicate a large contribution to the variance of theta * P6L11 "archives". Also this sentence is quite unclear and where my confusion about tau_a stems from * Eq 18 starts to get very confusing when the double square brackets are used to indicate expectations over different random variables. Perhaps use a subscript?

---

## Author Comment (AC1) · 18 Jun 2019

*The paper by Armhein addresses an important topic in palaeoclimatology, namely that of time representativeness of proxy data and the subsequent use of that data to compare between models and other data sets. I found the paper very hard to read and very technical, especially for COTP, though I would consider myself at the more technical end of the audience for this type of paper. I wonder if this work might be served better by publishing the mathematics in a more theoretical journal and subsequently interpretation and case studies in COTP.*

A key goal of this paper is to communicate some results from time series analysis, which give us a powerful set of tools for thinking about paleo sampling issues, to

the paleo community. I welcome the feedback given on how to streamline the presentation of the mathematical details so that the paper is well-suited to the readership of COTP.

*I found the figures well-presented but hard to interpret, and I found the table of notation (Table 1) particularly unhelpful; it looks like it was put together in 5 minutes.*

The next draft will clarify figure captions and interpretations.

*The key concept seems to revolve around the idea that there is a target measurement x which is desired to be averaged over temporal period $\tau_x$, but the scientist only has access to an observation y which is averaged over an interval of $\tau_y$. The origin of both the target and the observation is the 'true' time series $r(t)$. The paper's goal seems to be to estimate the error in using y to estimate $x$. I found this slightly strange, for two reasons. The first is that $x$ and $y$ are assumed to be centred at different time points. It's not clear to me why the centring needs to be (that) different.*

I included the possibility of an offset between x and y both to make the problem more general (as such offsets inevitably arise, and I was curious what their effect was) and to permit computing errors from age model uncertainty, which are common in paleoclimate and which are modeled here as a stochastic offset. The zero-offset instance is a specific case of the general treatment.

*The second is that, if I were doing this, I would try to estimate $r(t)$ as well as I possibly could, and then create smooths from this to allow me to compare with other time series. Perhaps this isn't possible, but I couldn't see why.*

It is challenging to estimate $r(t)$ (one has to use paleo data). In particular, we have effectively no information about $r(t)$ at frequencies higher than data sampling

rate, and yet this information is important for estimating representational errors. In the paper I argue that we don't need $r(t)$, but rather an estimate of its statistics, which simplifies the problem somewhat.

*I wonder if some of these problems might be solved by having a maximally simple running example at the start that makes clear the novelties of the approach using the bare minimum of maths/notation. That seems to be what has been attempted in Figure 1, though this just raises further questions. The light grey line appears to be $r(t)$ (this isn't mentioned?) which is unobserved. Labelling the vertical axis as $r(t)$ is slightly confusing since $y$ and $x$ are not values from r(t) but averages from it. We would like to get $x$ from $y$ where $y$ is centred on $t$ minus $\Delta$ over period $\tau_y$ and $x$ is centred on $t$ over period $\tau_x$. The key quantity theta is defined as the different between x and y, and most of the maths proceeds with the assumption that $r(t)$ is weakly stationary.*

Thank you for these comments for improving the figure and for the suggestion of a maximally simple running example at the start of the paper. I can see using such an example as a tool for introducing a (reduced) version of the mathematics, and including more of the technical details in the appendix.

*At this point I find myself a bit stuck as to the appropriateness of how this works. I don't think I've ever seen a situation similar to this. Aside from the stationarity assumption (which might hold locally?) in all the cases I've worked we have multiple $y$ observations covering the spanned period. (I suppose it might be more realistic if $\tau_x < \tau_y$ as often we're interested in estimating a climate value at a more precise time point than the observations).*

I agree that usually more than one observation is being used. One of the motivations for this work, which I can emphasize in the next draft, is this problem of computing mean values from multiple observations. The minimum variance solution for a mean

value depends not just on the values of the multiple observations, but also on their uncertainties (see e.g. Wunsch (2006), 2.7.5). The problem of determining uncertainties individually (what I do in the paper) is thus crucial for determining how measurements should be weighted to estimate time means and uncertainties of paleoclimate intervals.

*I'm further confused by the lower panel of Fig 1 which I initially assumed was the distri- bution of the error described by $\tau_x$ (they nearly match in magnitude) but is actually a completely different measurement uncertainty. I don't think that panel is helpful here. However, the legend points out that the uncertainty in theta is characterised by: sam- pling procedures, variability of $r$, archive smoothing, and chronological uncertainty. I totally agree with these fantastically important points. It's a strangely important sen- tence to appear in a figure caption. Having the lower panel display just one of these is confusing.*

I definitely see how that can be confusing. The motivation for the lower panel is to illustrate how age model uncertainty can come into play. I will make this point more explicit, and move this key sentence to a more prominent location.

*In Section 2 the paper dives into a lot of technical detail which I followed mathematically but quickly lost all the interpretation. I was hoping to pick things up again in Figure 2 which shows the frequency representation of the boxcar function at the top and the power transfer function at the bottom for assumed values of $\Delta$, $\tau_x$, and $\tau_y$. This seems to show which frequencies are contributing most to the TR error. However I've read the bullet points in Section 2.3 about 5 times now and I can't follow them. They talk about certain summary statistics which aren't shown in the Figure (226 and 1325 years?).*

Thank you for the feedback. I will move more of the technical description to the appendix to make less of a roadblock for the reader, and will make more explicit

connections between the summary statistics and the figure.

*Figure 3 rescues things a little bit by being a bit clearer but even then it doesn't intro- duce the 4 panels or the colours so I can only make good guesses as to what some these plots are showing. The top two panels are particularly helpful. Unfortunately without any further interpretation of Figures 2 and 3 I was completely lost beyond Sec- tion 3.1, which is a big shame. I really wanted to follow this.*

I'm glad you found the top two panels helpful, and will make sure that the other panels are explained too (a key point here is about the role of nonstationarity, which you mentioned above).

Subsequent sections explore how errors from different kinds of sampling proce- dures can build on one another and how they depend on the spectral properties of $r(t)$. I can smooth this transition - emphasizing connections to the simple model and examples - to help the reader connect the dots.

Some minor points:

*\* I got very confused about the role of $\tau_a$. I thought the archive smoothing was represented by $\tau_y$ as that is what we are observing?*

I will clarify this in the next draft. The archive smoothing is meant to stand in for processes (e.g. bioturbation in sediment cores) that can smooth a record prior to sampling it. Sampling also has a smoothing effect, so that a sampled record has potentially been smoothed twice.

*\* Section 2 I thought was mis-named. It's not really a statistical model in a gen- erative sense. It's more a collection of useful summary statistics that can indicate*

*problems in temporal representativeness. ***

I would argue that the model is in fact generative, in that it yields an estimate of errors due to representativeness.

*The glossary in Table 1 needs at least a sentence of interpreta- tion for the more complicated quantities. Saying e.g. $H(v)$ represents the power transfer function is fine, but what would be more useful is to say that high values of this for fre- quencies $v$ indicate a large contribution to the variance of theta ***

Thank you for this suggestion, which I would be happy to implement.

*P6L11 "archives". Also this sentence is quite unclear and where my confusion about $\tau_a$ stems from ***

I will clarify this in the next draft.

*Eq 18 starts to get very confusing when the double square brackets are used to indicate expectations over different random variables. Perhaps use a subscript?*

Good suggestion, thank you.

Cited: Wunsch, C. (2006). *Discrete inverse and state estimation problems: with geophysical fluid applications.* Cambridge University Press.

---

## Author Comment (AC2) · 18 Jun 2019

Author Response

R1 *This study addresses an important issue, which often is forgotten in the paleodata comparisons. Paleoclimatic measurements are usually dated and mostly dating uncertainties are communicated, too. However, depending on the archive and sample methods, measurements are often time averages, integrated over a specific period. This averaging period needs to be taken into account if these measurements are compared to simulations or other observations with different averaging periods. This study makes an important contribution to quantifying this error source.*

[Figure]

*The topic is highly relevant for Climate of the Past, ideas are novel and substantial conclusion reached. I found the figures, which introduce the basics concepts, very clear and helpful illustration. However, I assume the level of text and presented mathematical background goes beyond the average reader of this journal. Therefore, I suggest major revision, which should mainly simplify the entire text, to make it clearer and more accessible to a broad audience.*

I appreciate the comments of Reviewer 1 and their support for the relevance of the work. One of my goals in writing this paper is to make some of the machinery of time series analysis, which I found important and insightful for thinking about problems of what proxies represent, relevant and accessible to the paleo community, so I welcome ideas on how to improve this.

Reflecting suggestions from both reviewers, I plan to modify the paper to emphasize a simplified mathematical description of the errors, motivated by a minimal working example that will motivate key concepts. The bulk of the derivations, which I hope could be of use to the growing community doing this sort of uncertainty quantification, will be moved to the appendix.

*- Line 11: Many expressions have not been introduced, yet and are therefore not clear to the reader. What is the "target interval". Better avoid abbreviations like "tau" in the abstract.*

I agree that these abbreviations are unnecessary in the abstract, and will define the terms more clearly.

*- Line 13: What is meant by "archive smoothing" and "anti-aliasing"?*

Archive smoothing is defined (later in the paper) by noting that

"In many cases, paleoclimate archive are smoothed prior to processing by bio-turbation, diagenesis, residence times in karst systems upstream of speleothems, or other effects. These processes can be complex and non-constant in time..."

Clearly a similar definition would be appropriate in the abstract and introduction.

*- Page 2, lines 3ff.: I would suggest bullet points for the various error sources*

Thank you, good idea.

*Fig. 1: Great, this makes the problem easily accessible.*

*Table 1: This looks more like the variable list of the 500-page book than for a CP article. It may help to have reduced (basic) version of the mathematical background in the main part of the paper and the derivations in a supplement.*

I would be glad to shift the balance towards the less technical in the main text to improve accessibility, and move this table to an appendix.

*Page 6, line 1: "boxcar" needs to be explained*
*Page 6, line 6: over over*
*Fig. 3: Labeling too small*
*References: Latex code remained in the pdf version.*

Thanks for these additional points, which I am happy to address.

---

## Author Response (AR1)

**Response to Reviewers for "How large are temporal representativeness errors in paleoclimatology?"**

**Dan Amrhein**

**August 31, 2019**

I appreciate the thoughtful reviews on this manuscript. I have addressed them point-by-point below, with reference to changes in the paper when appropriate.

**Reviewer 1**

*This study addresses an important issue, which often is forgotten in the paleodata comparisons. Paleoclimatic measurements are usually dated and mostly dating uncertainties are communicated, too. However, depending on the archive and sample methods, measurements are often time averages, integrated over a specific period. This averaging period needs to be taken into account if these measurements are compared to simulations or other observations with different averaging periods. This study makes an important contribution to quantifying this error source.*

*The topic is highly relevant for Climate of the Past, ideas are novel and substantial conclusion reached. I found the figures, which introduce the basics concepts, very clear and helpful illustration. However, I assume the level of text and presented mathematical background goes beyond the average reader of this journal. Therefore, I suggest major revision, which should mainly simplify the entire text, to make it clearer and more accessible to a broad audience.*

I appreciate the comments of Reviewer 1 and their support for the relevance of the work. One of my goals in writing this paper is to make some of the machinery of time series analysis, which I found important and insightful for thinking about problems of what proxies represent, relevant and accessible to the paleo community, so I welcome ideas on how to improve this.

Reflecting suggestions from both reviewers, I have overhauled the paper to emphasize a simplified description of the errors motivated by a minimal working example (Fig. 1). The bulk of the derivations have been moved to the appendix

and the number of equations in the main text reduced from 28 to 7.

*- Line 11: Many expressions have not been introduced, yet and are therefore not clear to the reader. What is the "target interval". Better avoid abbreviations like "tau" in the abstract.*

This term has been replaced by "climate interval of interest," and the abbreviations removed from the abstract.

*- Line 13: What is meant by "archive smoothing" and "anti-aliasing"?*

I have replaced "archive sampling" with an explanation that "the paleoclimate archive has been smoothed in time prior to sampling" and saved discussions of anti-aliasing filters until later.

*- Page 2, lines 3ff.: I would suggest bullet points for the various error sources*

Done, see new page 3.

*Fig. 1: Great, this makes the problem easily accessible.*

*Table 1: This looks more like the variable list of the 500-page book than for a CP article. It may help to have reduced (basic) version of the mathematical background in the main part of the paper and the derivations in a supplement.*

As suggested, the great bulk of the derivations was moved to the appendix.

*Page 6, line 1: "boxcar" needs to be explained*

This content has been moved to Appendix A, where it is explicitly defined.

*Page 6, line 6: over over*

Fixed.

*Fig. 3: Labeling too small*

Thank you, I oriented the figure in landscape to make fonts larger.

*References: Latex code remained in the pdf version.*

Fixed, thank you.

**Reviewer 2**

*The paper by Armhein addresses an important topic in palaeoclimatology,*

*namely that of time representativeness of proxy data and the subsequent use of that data to compare between models and other data sets. I found the paper very hard to read and very technical, especially for COTP, though I would consider myself at the more technical end of the audience for this type of paper. I wonder if this work might be served better by publishing the mathematics in a more theoretical journal and subsequently interpretation and case studies in COTP.*

A key goal of this paper is to communicate some results from time series analysis, which give us a powerful set of tools for thinking about paleo sampling issues, to the paleo community. I welcome the feedback given on how to streamline the presentation of the mathematical details so that the paper is well-suited to the readership of COTP. As part of restructuring the paper, I moved most of the equations and technical details to the appendices and tried to focus on results that are useful for constructing and interpreting paleo records.

*I found the figures well-presented but hard to interpret, and I found the table of notation (Table 1) particularly unhelpful; it looks like it was put together in 5 minutes.*

I worked to streamline figure captions and eliminated the table, reflecting the simpler descriptions now in the main text.

*The key concept seems to revolve around the idea that there is a target measurement x which is desired to be averaged over temporal period $\tau_x$, but the scientist only has access to an observation y which is averaged over an interval of $\tau_y$. The origin of both the target and the observation is the 'true' time series $r(t)$. The paper's goal seems to be to estimate the error in using y to estimate x. I found this slightly strange, for two reasons. The first is that x and y are assumed to be centred at different time points. It's not clear to me why the centring needs to be (that) different.*

I included the possibility of an offset between x and y both to make the problem more general (as such offsets inevitably arise, and I was curious what their effect was) and to permit computing errors from age model uncertainty, which are common in paleoclimate and which are modeled here as a stochastic offset. The zero-offset instance is a specific case of the general treatment.

*The second is that, if I were doing this, I would try to estimate $r(t)$ as well as I possibly could, and then create smooths from this to allow me to compare with other time series. Perhaps this isn't possible, but I couldn't see why.*

It is challenging to estimate $r(t)$ (one has to use paleo data). In particular, we have effectively no information about $r(t)$ at frequencies higher than data sampling rate, and yet this information is important for estimating representational errors. In the paper I argue that we don't need $r(t)$, but rather an estimate of its statistics, which simplifies the problem somewhat. This point is

now discussed starting on p3 l14.

*I wonder if some of these problems might be solved by having a maximally simple running example at the start that makes clear the novelties of the approach using the bare minimum of maths/notation. That seems to be what has been attempted in Figure 1, though this just raises further questions. The light grey line appears to be $r(t)$ (this isn't mentioned?) which is unobserved. Labelling the vertical axis as $r(t)$ is slightly confusing since $y$ and $x$ are not values from r(t) but averages from it. We would like to get $x$ from $y$ where $y$ is centred on $t$ minus $\Delta$ over period $\tau_y$ and $x$ is centred on $t$ over period $\tau_x$. The key quantity theta is defined as the different between $x$ and $y$, and most of the maths proceeds with the assumption that $r(t)$ is weakly stationary.*

Figure 1 has been expanded to give more intuition for different kinds of error that may arise and to serve as a more complete maximally simple example. I refer back to it repeatedly in the text.

*At this point I find myself a bit stuck as to the appropriateness of how this works. I don't think I've ever seen a situation similar to this. Aside from the stationarity assumption (which might hold locally?) in all the cases I've worked we have multiple $y$ observations covering the spanned period. (I suppose it might be more realistic if $\tau_x < \tau_y$ as often we're interested in estimating a climate value at a more precise time point than the observations).*

I agree that usually more than one observation is being used. One of the motivations for this work, which I have emphasized here (p7, l13), is this problem of computing mean values from multiple observations. The minimum variance solution for a mean value depends not just on the values of the multiple observations, but also on their uncertainties (see e.g. Wunsch (2006), 2.7.5). The problem of determining uncertainties individually (what I do in the paper) is thus crucial for determining how measurements should be weighted to estimate time means and uncertainties of paleoclimate intervals.

*I'm further confused by the lower panel of Fig 1 which I initially assumed was the distribution of the error described by $\tau_x$ (they nearly match in magnitude) but is actually a completely different measurement uncertainty. I don't think that panel is helpful here. However, the legend points out that the uncertainty in theta is characterised by: sam- pling procedures, variability of $r$, archive smoothing, and chronological uncertainty. I totally agree with these fantastically important points. It's a strangely important sen- tence to appear in a figure caption. Having the lower panel display just one of these is confusing.*

Figure 1 has been modified to remove the chronological uncertainty PDF. A sentence describing the different origins of uncertainty appears now in the abstract (l. 9).

*In Section 2 the paper dives into a lot of technical detail which I followed mathematically but quickly lost all the interpretation. I was hoping to pick things up again in Figure 2 which shows the frequency representation of the boxcar function at the top and the power transfer function at the bottom for assumed values of $\Delta$, $\tau_x$, and $\tau_y$. This seems to show which frequencies are contributing most to the TR error. However I've read the bullet points in Section 2.3 about 5 times now and I can't follow them. They talk about certain summary statistics which aren't shown in the Figure (226 and 1325 years?).*

Thank you for the feedback. I have moved the technical description to the appendix to make less of a roadblock for the reader. Rather than describing explicitly the cutoff frequencies of the transfer function, I discuss qualitatively the fact that one frequency band is primarily responsible for error variance.

*Figure 3 rescues things a little bit by being a bit clearer but even then it doesn't intro- duce the 4 panels or the colours so I can only make good guesses as to what some these plots are showing. The top two panels are particularly helpful. Unfortunately without any further interpretation of Figures 2 and 3 I was completely lost beyond Sec- tion 3.1, which is a big shame. I really wanted to follow this.*

A more detailed description of all of the panels in this figure now appears on p7, l26. I have worked to smooth the transition between the technical setup and the subsequent interpretation to help the reader connect the dots. Hopefully the paper is now easier to follow.
Some minor points:

*\* I got very confused about the role of $\tau_a$. I thought the archive smoothing was represented by $\tau_y$ as that is what we are observing?*

The archive smoothing (now defined on p3, l25) is meant to stand in for processes (e.g. bioturbation in sediment cores) that can smooth a record prior to sampling it. Sampling also has a smoothing effect, so that a sampled record has potentially been smoothed twice.

*\* Section 2 I thought was mis-named. It's not really a statistical model in a generative sense. It's more a collection of useful summary statistics that can indicate problems in temporal representativeness. \**

In the new version, this (now less technical) section has been renamed "Origins of temporal representativeness error"

*The glossary in Table 1 needs at least a sentence of interpreta- tion for the more complicated quantities. Saying e.g. $H(v)$ represents the power transfer function is fine, but what would be more useful is to say that high values of this for fre- quencies $v$ indicate a large contribution to the variance of theta \**

Thank you for this suggestion. This table has been removed.

*P6L11 "archives". Also this sentence is quite unclear and where my confusion about $\tau_a$ stems from *

This sentence has been removed and archive sampling defined on p3, l25.

*Eq 18 starts to get very confusing when the double square brackets are used to indicate expectations over different random variables. Perhaps use a subscript?*

Done; thank you.

Cited: Wunsch, C. (2006). *Discrete inverse and state estimation problems: with geophysical fluid applications.* Cambridge University Press.

[revised manuscript text omitted]

where $\nu$ denotes frequency, $H$ is a so-called transfer function, and $|\hat{r}(\nu)|^2$ is the power spectral density of the true signal $r(t)$. In effect, the error variance is a weighted sum of the power at different frequencies in $r(t)$, where the weights in frequency space (given by $H$) depend on how the paleoclimate record has been sampled and smoothed. This behavior is typical of aliasing,

where variance in the signal at one frequency appears erroneously in a measurement at a different frequency (in this case, at the zero frequency, which is the time mean).

[Figure]

**Figure 2.** The power transfer function $H$ (Equation 3) illustrates the dependence of temporal representativeness errors on frequencies in the climate signal and on sampling time scales. In the case where the offset $\Delta$ between measurement and target is 0, $H$ is a squared difference of sinc functions $\mathrm{sinc}(\tau \nu)$ (A13), illustrated here for $\tau = 100$ years and $\tau = 1000$ years. (b) Transfer functions for three different values of the time offset $\Delta$. Grey bars indicate the 1/200 and 1/1300 yr$^{-1}$ frequencies, which approximately bound the frequencies contributing to TR errors in the $\Delta = 0$ case.

While the details are left to the appendix, it is noteworthy that in many practical cases, TR errors can be straightforwardly attributed to signal variability within a particular frequency band. This frequency-band behavior emerges because $H$ is a squared
5  difference of sinc functions (Figure 2a), which has a bump-like shape (Figure 2b). For instance, if a centennial mean is used to represent a millennial mean, in the absence of archive smoothing, the expected error variance is roughly equal to the variance in $r(t)$ at periods between 200 and 1300 years; the error is the same if a centennial mean is used to represent a decadal mean. Thus the difference between the sample and target averaging intervals ($\tau_y$ and $\tau_x$) sets the frequency band that is aliased onto the mean. These effects are modulated in the presence of archive smoothing, and when there is a time offset in the measurement
10  relative to the target, additional variability is aliased onto errors.

**4  Illustrating TR error quantification at the Last Glacial Maximum by sampling a high-resolution paleoclimate archive**

Here we explore the procedure for estimating TR errors described in the previous section in the context of estimating mean properties at the Last Glacial Maximum (LGM), the period roughly 20,000 years ago that is associated with the greatest land ice extent during the last glacial period. Following MARGO Project Members (2009) and others, LGM target quantities are defined to be estimates of time means over the 4000-year-long period from 23,000 to 19,000 years ago (23-19 kya),

$$x_{LGM} = \frac{1}{4000} \int_{-23,000}^{-19,000} r(t)\,dt.$$

We will consider TR errors arising from representing $x_LGM$ by an observed 1000-year time-mean value that is centered on 21 kya,

$$y_{LGM} = \frac{1}{1000} \int_{-21,500}^{-20,500} r(t)\,dt. \tag{4}$$

Qualitatively, errors from this representation have the form illustrated in Figure 1a. Such an estimate – dated to within the LGM, but averaging over only a subset – could reasonably be included in a binned-average compilation of LGM data. However, because statistically robust averaging procedures must downweight uncertain observations according to observational error, including TR errors, is important to avoid biasing any binned averages. Similarly, were we to compare $y_{LGM}$ to an LGM-mean estimate of $r(t)$ from a model without taking TR errors into account, we might erroneously conclude that the model did not fit the data.

How large is the TR error in representing $x_{LGM}$ by $y_{LGM}$? We will illustrate the procedure proposed in Section 3 by taking a high-resolution climate record to be a true climate signal $r(t)$ and sampling it at longer time averages than the record spacing. Here we will use the the North Greenland Ice Core Project (NGRIP; Andersen et al. (2004)) 50-year average time series of oxygen isotope ratios ($\delta^{18}$O). Equation 2 states that TR error variance is equal to the squared difference between running means of $r(t)$, averaged over the record length. Smoothing the NGRIP record with running means of length $\tau_x = 4000$ and $\tau_y = 1000$ yields time series of potential target and observation values $x$ and $y$ (black and red lines, Figure 3a). Their difference is the error $\theta$ (thick black line, 3a), and their squared difference is the blue line in 3b. The time mean $\langle \theta^2 \rangle$ (red line, Figure 3b) is 0.7 $\left(\text{‰}\delta^{18}\text{O}\right)^2$ and is the estimate of the error variance.

A prominent feature in Figure 3b is the time variability of TR error: in some time periods (including the LGM) errors are relatively small, whereas they are markedly larger at, e.g., 80-70 kya. This time-variability in errors arises from nonstationarities in the NGRIP oxygen isotope record. The transfer function (blue line, 3c), shows that for our choices of $\tau_x$ and $\tau_y$, variability in the frequency band lying between roughly 2200 and 5300 year periods is responsible for TR errors. A wavelet analysis (Figure 3d) shows that increased variability in this band is coincident with increases in TR error variance: note, e.g., the the correspondence of high wavelet values in that band near -75000 years with contemporaneous large values in the blue line in 3b. Evidently, in the presence of nonstationary climate variability, TR errors can vary in time. They may also vary due to changes

[Figure]

**Figure 3. Temporal representativeness error in the time and frequency domains.** Errors in representing a 4000-year mean by a 1000-year mean are estimated by computing the difference $\theta$ ((a), thick black line) between a 4000-year (red line) and 1000-year (thin black line) running mean of the NGRIP $\delta^{18}O_{ice}$ record (grey). The time average (red line, (b)) of $\theta^2$ (blue line) is an estimate (0.7, units of $\left(\%_0 \delta^{18}O\right)^2$) of the temporal representativeness error variance. Large values in $\theta^2$ correspond to time periods with increased variability, as diagnosed by a wavelet analysis (d), particularly in the band between 2257 and 5298 year periods (grey lines). The light blue line in panel (c) is the power spectrum $\nu^{-\beta}$ with $\beta = 1.53$ derived by a least-squares fit to the NGRIP spectrum.

in sampling procedures over the course of constructing a time series, as discussed in Section 6. Observations of intervals with less variability in the TR error frequency band (e.g., the LGM) will be less susceptible to TR errors, an additional quantitative justification for the long-held process of focusing study on time-means of periods with relatively less variability.

[Figure]

**Figure 4.** Error-to-signal variance fractions $f$ (Equation 5) for estimates of time-mean values computed from the NGRIP record of Pleistocene oxygen isotopes contoured as a function of target averaging interval $\tau_x$ and observation averaging interval $\tau_y$. A value of 0.1 means that TR error amplitudes are 10% of the "signal," which defined as the typical difference between two time averages over durations $\tau_x$ separated by 21,000 years.

Next, we will extend our analysis of NGRIP to cover a range of different values of $\tau_x$ and $\tau_y$. To compare the NGRIP results
5  to synthetic time series in the following sections with arbitrary units, we will analyze the unitless noise-to-signal standard deviation ratio,

$$f = \frac{\sqrt{\langle \theta^2 \rangle}}{\sigma}. \tag{5}$$

Because one motivation of studying the LGM is inferring differences from modern climate, we adopt as our "signal" amplitude the typical anomaly $\sigma$ between two mean intervals of length $\tau_x$ separated by $= 21,000$. This quantity is estimated from the
10  NGRIP time series for each value of $\tau_x$.

Recall that for $\tau_x = 4000$ and $\tau_y = 1000$, the estimated error variance was $0.7\ \left(‰\delta^{18}O\right)^2$. Figure 4 contours the results of the same calculation (now expressed as the noise-to-signal ratio $f$) for every combination of $\tau_x$ and $\tau_y$ between 10 and 4000 years. TR errors depend jointly on values of $\tau_x$ and $\tau_y$. Errors are zero for $\tau_x = \tau_y$ (corresponding to an ideal sampling scheme) and increase monotonically away from those values. Errors are greatest (up to 30% of signal amplitudes) for small values of
15  $\tau_y$ relative to $\tau_x$, where TR error dwarfs the relatively small signal amplitudes that are typical of 21,000-year differences in

long-term time averages[1]. Subsequent sections extend this analysis to a broader set of sampling parameters (including archive smoothing and time offsets) as well as records with different spectral characteristics.

**5 Exploring interactions between sampling parameters and signal spectra**

The succinct expression of TR errors in terms of power spectra (3) is a clue that the spectral character of paleoclimate processes are an important factor for the amplitude of TR errors. To investigate how errors depend on the spectrum of $r(t)$, we will shift our focus away from observations and consider climate processes with power-law spectra, i.e. those whose power spectral densities $|\hat{r}(\nu)|^2$ have the form

$$|\hat{r}(\nu)|^2 \propto \nu^{-\beta}, \tag{6}$$

where $\beta$ is termed the spectral slope (when plotted in log-log space, $\nu^{-\beta}$ is a straight line with slope $-\beta$). We choose this idealized form because spectra consistent with a power-law description are common in climate (Wunsch, 2003). White noise, which partitions variance equally across frequencies, has a spectral slope of 0; signals with a steeper slope (larger $\beta$) have a larger fraction of their variance originating from low-frequency variability. Here we consider spectral slopes $\beta = 0.5$ and $\beta = 1.5$, motivated by Huybers and Curry (2006), who fit paleoclimate records to spectral slopes between $\beta = 0.3$ and $\beta = 1.6$. Climatological spectral features that are not described by power laws, such as peaks due to deterministic astronomical forcing from annual or Milankovich variability, also contribute to errors (Pisias and Mix, 1988; Wunsch, 2000) but are not considered specifically in these examples. All calculations are performed by numerical integration of Equation (A13) by global adaptive quadrature.

**5.1 Effects from archive smoothing and spectra**

Similar to Figure 4, Figure 5 contours the noise-to-signal ratio $f$ as a function of $\tau_x$ and $\tau_y$, but now for four cases spanning two values of the archive smoothing time scale $\tau_a$ (0 and 1000 years) and two values of spectral slope $\beta$. Signals with steeper spectral slopes ($\beta = 1.5$ rather than $\beta = 0.5$), show smaller $f$ values because TR errors, which originate at relatively high frequencies (Figure 2), are smaller relative to the greater amount of low-frequency variability, as discussed also by Wunsch (1978) and Wunsch (2003). The close resemblance between Figure 5b the equivalent figure computed from NGRIP (4), which has spectral slope of 1.53 (Figure 3c), is partly coincidental; analysis of synthetic records with spectral slopes of 1.5 (not shown) reveals variability in $f$ because of variations about the power law distribution in finite-length, stochastically generated time series, of which NGRIP is arguably one realization. Nevertheless, the agreement shows correspondence between results from paleoclimate data and our idealized approach.

Dependencies on $\tau_x$ and $\tau_y$ change when we include archive smoothing (Figures 5a and 5; schematized in Figure 1a and 1c). These effects are evident primarily for $\tau_y < \tau_a$. In that "smoothed" regime, the largest values of $f$ for small $\tau_y$ are reduced because archive smoothing removes some of the high-frequency variability that would otherwise be felt by observations and
* * *
[1]Error variances are equal if $\tau_x$ and $\tau_y$ are interchanged, but asymmetry in $f$ arises because $\sigma$ depends on $\tau_x$.

[Figure]

**Figure 5.** Error-to-signal fractions $f$ for time-mean estimates plotted as a function of target averaging interval $\tau_x$ and observation averaging interval $\tau_y$. Climate signal spectra are approximated as power law functions of frequency ($|\hat{r}(\nu)|^2 \propto \nu^{-\beta}$) with spectral slopes $\beta$ equal to 0.5 (left column) and 1.5 (right column). The top row corresponds to a case with no archive smoothing ($\tau_a = 0$) while the bottom row corresponds to a case where the signal $r(t)$ is smoothed by a running mean over $\tau_a = 1000$ years. Values to the left of the bold line at $\tau_y = \tau_a$ lie in the "smoothed regime" where archive smoothing qualitatively affects results. Time scales were chosen to be relevant to the problem of time-mean estimation at the Last Glacial Maximum, ca. 20 kya. Dotted lines show values of $\tilde{\tau}_y$ derived to minimize error estimated according to Equation (A20).

erroneously aliased onto the mean. Another effect is that $\tau_y = \tau_x$ no longer minimizes $f$ everywhere; in the smoothed regime, smaller values of $\tau_y$ lead to reduced TR error. This is because archive smoothing already provides a measure of time averaging; so that when $\tau_x = 1000$, the value of $\tau_y$ that minimizes error is close to zero, because anything longer would be "oversmoothing" the record and effectively giving a longer time average than $\tau_x$. Archive smoothing also reduces the sensitivity of errors to the

5 choice of $\tau_y$ for $\tau_y < \tau_a$. Finally, the presence of smoothing means that arbitrarily short choices of $\tau_x$ can no longer be resolved without error, as evidenced by the monotonic growth of error as $\tau_x$ decreases from $\tau_a$.

To the extent that these simple experiments reflect actual paleoclimate sampling procedures, one could attempt to sample time-mean intervals to avoid TR errors. In the absence of archive sampling, the (trivial) result is that $\tau_y$ should be equal to $\tau_x$. But the danger of oversmoothing means that this rule is not always appropriate for $\tau_a \neq 0$. Appendix A derives (A20) an

10 approximate expression for the error-minimizing value, $\tilde{\tau}_y = \sqrt{\tau_x^2 - \tau_a^2}$, that is a function of both the target interval length and smoothing time scale. These values (dotted lines, Figure 5) are in good qualitative agreement with minimum TR error values.

**5.2 Effects from known time offsets**

Having explored how choices of $\tau_x$ and $\tau_y$ contribute to TR errors, we next illustrate effects from chronological offsets when $\Delta \neq 0$ (schematized in 1b). Motivated by the LGM time scale, we focus again on the case where $\tau_x$ is fixed to 4000 years

15 and integrate A13 varying $\tau_y$ between 10 and 6000 years and $\Delta$ between 10 and 4000 years for values of $\beta = (0.5, 1.5)$ and $\tau_a = (0, 1000)$.

For all values of $\tau_a$ and $\beta$, errors grow monotonically away from the values $\Delta = 0$, $\tau_y = \tau_x$, which corresponds to the case with no TR error[2] (Figure 6). As in the previous section, a "smoothed regime" is evident for $\tau_y \leq \tau_a$ across all values of $\Delta$ shown: because archive smoothing damps variability in time, the errors from shifting an observation relative to the target

20 become less severe. Another qualitative difference emerges for values of $\Delta$ that are greater or less than $|\tau_x - \tau_y|/2$ (blue line, Figure 6a and 6c). This boundary designates when the observed time period is sufficiently offset that it begins to fall outside the target interval; at that point, errors grow rapidly as $\Delta$ increases. As before, errors are more pronounced for $\beta = 0.5$ than for $\beta = 1.5$, with errors larger than the signal ($f > 1$) for small values of $\tau_y$ at all values of $\Delta$ for $\beta = 0.5$.

**5.3 Effects from probabilistic time offsets**

25 When the dating of a measurement is uncertain, a range of $\Delta$ values may be possible, as specified by a probability distribution function $p(\Delta)$. To explore effects from chronological uncertainty, we assume that $p(\Delta)$ is Gaussian about zero with standard deviation equal to the time scale $\sigma_\Delta$. We include this uncertainty in TR error variance by taking a second expectation (denoted by $\langle\rangle_\Delta$, in addition to the time expectation in Equation 2) to give

$$\langle\langle\theta^2\rangle\rangle_\Delta = \int_{-\infty}^{\infty} p(\Delta)\langle\theta^2\rangle\, d\Delta. \tag{7}$$
* * *
[2]A small amount of oversmoothing is present at $\tau_y = \tau_x$ in the $\tau_a = 1000$ case, but is not qualitatively important.

[Figure]

**Figure 6.** Same as Figure 5, but illustrating effects of offsets $\Delta$ between target and observational intervals on noise-to-signal ratios. Error fractions $f$ are plotted as a function of the observational averaging interval $\tau_y$ and the standard deviation $\sigma_\Delta$ of a Gaussian distribution of observational offset centered on zero. In all cases, the target averaging interval is $\tau_x = 4000$, reflecting the nominal length of the Last Glacial Maximum. Values along the line $\tau_y = \tau_x$ strictly reflect the influence of chronological offsets. The blue line in panel (a) denotes values for which $\Delta = \left| \tau_x - \tau_y \right| / 2$, indicating the maximum values of $\Delta$ for which $\tau_x$ and $\tau_y$ completely overlap.

[Figure]

**Figure 7.** Same as Figure 5, but illustrating effects of chronological uncertainties in observations on noise-to-signal ratios. Error fractions $f$ are plotted as a function of the observational averaging interval $\tau_y$ and the standard deviation $\sigma_\Delta$ of a Gaussian distribution of time offsets centered on zero. In all cases, the target averaging interval is $\tau_x = 4000$, reflecting the nominal length of the Last Glacial Maximum. Values along the line $\tau_y = \tau_x$ strictly reflect the influence of chronological uncertainty, which is zero when the observational offset is exactly known to be zero, (i.e., $\sigma_\Delta = 0$).

In practice, $p(\Delta)$ can be multimodal or otherwise non-Gaussian (e.g., from radiocarbon ages; Telford et al. (2004)) which could qualitatively change results. While not explored here, such errors can be investigated by integrating Equation (7) with different choices of $p(\Delta)$.

Integrating (7) and varying $\sigma_\Delta$ and $\tau_y$ shows that TR errors in the presence of Gaussian chronological uncertainty $p(\Delta)$
5   with standard deviation $\sigma_\Delta$ are qualitatively similar to those from a fixed offset $\Delta = \sigma_\Delta$ (cf. Figures 6 and 7). The transition in sensitivity to $\sigma_\Delta$ across $\sigma_\Delta = |\tau_x - \tau_y|/2$ is less pronounced than for the equivalent in Figure 6, consistent with the range of lags that is possible for any nonzero $\sigma_\Delta$. Nevertheless, the intuitively sensible conclusion is that chronological errors will be gravest when uncertainties tend to place measurements outside of target intervals. Reduced errors in the smoothed regime $\tau_y \leq \tau_a$ indicate that archive smoothing can reduce effects from chronological errors in some cases.

10   ## 6   Extension to time series

Paleoclimate time series are sequences of time-mean values; here, we discuss how the TR errors discussed for time-mean estimation affect transient records of climate variability. We show that in the absence of archive smoothing, dense sampling (i.e., setting the averaging interval equal to the spacing between measurements) is a nearly optimal approach to minimize TR errors.

15   The sampling theorem of Shannon (1949) states that sampling $r(t)$ instantaneously at times separated by a fixed time interval $\tau_s$ unambiguously preserves signal information only when $r(t)$ does not contain any spectral power at frequencies greater than $1/2\tau_s$ (called the Nyquist frequency, $\nu_{Nyq}$). When this criterion is not met, the discrete signal is corrupted by aliasing, whereby variability in $r(t)$ at frequencies greater than $\nu_{Nyq}$ appears artificially at lower frequencies in the discrete signal. To mitigate aliasing, one can either increase the sampling rate or apply a low-pass "anti-aliasing" filter to $r(t)$ to attenuate power at
20   frequencies higher than $\nu_{Nyq}$.

In the process of constructing a paleoclimate time series, sampling time-mean values serves a moving average and, thereby, an anti-aliasing filter. Thus we expect sample averaging procedures to affect aliasing errors in time series, as also discussed by von Albedyll et al. (2017). Appendix B uses Shannon's theorem to obtain a frequency-domain expression for the TR errors for individual time series measurements. The procedure is to 1) define local (in time) values of $\tau_s^i$ and $\nu_{Nyq}^i$ for the $i^{th}$ observation
25   and 2) compute the expected errors if an entire time series were sampled using those local properties. To do this, we make the assumption that the sampling interval $\tau_s^i$ is locally constant: that is, for the $i^{th}$ measurement $y^i$ taken at time $t^i$, $y^{i-1}$ was taken at time $t^i - \tau_s^i$, and $y^{i+1}$ was taken at time $t^i + \tau_s^i$. If the sampling interval changes rapidly, conclusions from this approach might not apply. Again leaving the details to the appendix, we note that similar to the time-mean case, the error variance $\langle \theta^{i2} \rangle$ for the $i^{th}$ observation is a weighted integral over the power density spectrum of $r(t)$ (B6). Unlike in the mean estimation case,
30   where TR errors can be zero, nonzero error is unavoidable with uniform sampling because of differences between the shape of the sinc function and the ideal, abrupt frequency cutoff that minimizes error according to Shannon's theorem.[3]
* * *
[3]Sampling a paleoclimate archive nonuniformly in time could better approximate the ideal filter and reduce errors, but this may not be practical given the challenges of recovering and sampling paleoclimate data.

To demonstrate sensitivities to sampling parameters we again compute noise-to-signal ratios. In keeping with our local measure of TR error, we take the signal strength to be the standard deviation of the time series that would result if $r(t)$ were sampled with the same averaging and sampling interval as the $i^{th}$ observation over 21,000 years, the approximate duration of the last deglaciation. The results are qualitatively similar to those for the time-mean case, with two main distinctions (cf.

5   Figure 8 and 5). First, as discussed above, errors are always 10% or more of signal amplitudes because of errors arising from constructing a time series as a sequence of time mean values. Second, values of $\tau_y$ that minimize errors do not obey $\tau_y = \tau_s$, but are larger by a factor of roughly 1.2, suggesting that, absent considerations from archive smoothing, the ideal sample would span an interval slightly longer than the sampling interval to minimize errors. In practice, sampling densely (that is, without space between observations) is a good approximation of this error minimizing strategy.

10   As in the time-mean case, the effects of archive smoothing are large in a regime of sampling parameter space ($\tau_y \leq \tau_a$), implying that knowledge of $\tau_a$ is important for informing choices of $\tau_s$ and $\tau_y$. Clearly, sampling at intervals $\tau_s < \tau_a$ will result in errors because some of the variability of interest will have been destroyed. Choosing a $\tau_s$ that is larger than both $\tau_y$ and $\tau_a$ will result in aliasing errors. Finally, within the smoothed regime $\tau_y \leq \tau_a$, TR errors are less sensitive to choices of $\tau_y$ than they are for $\tau_y \geq \tau_a$, meaning that sampling discontinuously (i.e., not densely) may not be problematic.

15  **7   Discussion**

This paper presents a framework for quantifying temporal representativeness (TR) errors in paleoclimate, broadly defined as resulting when one time average is represented by another. A simple model illustrates interacting effects from record sampling procedures, chronological errors, and the spectral properties of the climate process being sampled.

We find that TR errors for time mean estimates can be large relative to climate signals, with noise-to-signal standard deviation

20   ratios greater than 1 in some cases, particularly where the observational interval $\tau_y$ is smaller than the target interval $\tau_x$. These errors result from aliasing climate variability onto time mean estimates and can be mitigated to some degree by choices of sampling procedures and by archive smoothing, both of which act as anti-aliasing filters. However, archive smoothing can also destroy information about climate variability, and the combined effects of sampling and smoothing can over-smooth a record and lead to increased errors. TR errors due to sampling are not independent of chronological errors, but interact, for instance in

25   the way that errors grow more quickly as a function of chronological uncertainty amplitude when that uncertainty is likely to place a measurement outside of a target interval (Figure 7). Given that these error variances were estimated using parameters representative of the LGM, it seems possible that TR errors may explain some of the disagreement among proxy measurements within that time period (e.g., MARGO Project Members, 2009; Caley et al., 2014), though nonstationarities may cause TR errors to be overestimated for climate intervals like the LGM that appear to be quiescent relative to other time periods. While

30   we do not claim that TR errors are the largest source of error for any particular proxy type or reconstruction problem, they may be in some cases. The tools presented can be used to assess likely error amplitudes.

Though not the principal goal, these analyses provide a basis for sampling practices that minimize errors, for instance for avoiding oversmoothing that can arise through the combined effects of sampling and archive smoothing. When constructing

[Figure]

**Figure 8.** Same as Figure 5, but illustrating the dependence of the error-to-signal standard deviation ratio for individual measurements in a time series as a function of local time series spacing ($\tau_s^i$) and the observational averaging time interval $\tau_y^i$.

paleoclimate time series, it is important to bear in mind not just the Nyquist frequency but the role of sampling and smoothing time scales as anti-aliasing filters; these considerations point to dense sampling (i.e., without space between contiguous samples) in order to minimize error in the absence of effects from archive smoothing (Section 6). However, many practical considerations motivate paleoclimate sampling strategies, and may outweigh the concerns described here. For instance,

5   records sampled densely cannot be used as a starting point for subsequently constructing higher-resolution records. Moreover, preservation of natural archives for subsequent analyses is important for reproducibility and for sharing resources between laboratories, and may be complicated by continuous sampling.

To some extent, the simple model for TR error can be generalized to more complex scenarios. If samples are nonuniform in time – for instance, due to large changes in chronology, or because material was sampled using a syringe or drill bit with

10   a circular projection onto an archive – then the sinc function in (3) can be replaced by Fourier transforms of the relevant function. Similarly, a more complex pattern of archive smoothing can be accommodated by substituting a different smoothing kernel. Non-Gaussian age uncertainties can be incorporated by substituting a different distribution in (7). Changes in sampling properties through time (as might arise from non-constant chronologies or sampling procedures) can readily be accommodated because all computations are performed on a point-by-point basis. If sampling or smoothing time scales are unknown, a similar

15   procedure can be adopted as was used for $\Delta$ in (7), whereby a second integration is performed to compute the expectation over an estimated probability distribution of one or more time scales.

Several caveats apply to the uncertainty estimates given. First, the model neglects some effects that may be important, such as inhomogeneities in preserved climate signals owing to e.g. diagenesis or scarcity of biological fossils. Second, nonstationarity in record spectra leads to time variations in errors, as illustrated in Figure 3. Third, in the analysis of time series errors, we

20   ignore the possibility that errors covary in time, which can result from chronologies constructed by interpolating ages between tie points; more complete characterizations could be achieved by Monte Carlo sampling of age model uncertainty (Anchukaitis and Tierney, 2013). More broadly, there is clear need for comprehensive approaches in uncertainty quantification that can reveal interactions among the various sources of uncertainty in paleoclimate records. Forward proxy system models (e.g., Evans et al., 2013; Dee et al., 2015; Dolman and Laepple, 2018) are a promising way forward to assess uncertainties holistically.

25   Results for time series (Section 6) hold when record spacing and chronologies do not change too rapidly and where the goal is to obtain a discrete representation of a continuous process. For other objectives, other sampling procedures may be preferred. For instance, "burst sampling," whereby rapid sequences of observations are taken at relatively long intervals, is used in modern oceanographic procedures to estimate spectral nonstationarities in time (Emery and Thomson, 2014), and unevenly spaced paleoclimate observations can be leveraged to give a range of frequency information using variogram approaches (Amrhein

30   et al., 2015) or the Lomb-Scargle periodogram (e.g., Schulz and Stattegger, 1997).

Representativity errors due to aliasing are not limited to the time domain, and similar procedures may be useful for quantifying errors due to spatial representativeness by considering how well proxy records can constrain the regional and larger scales typically of interest in paleoclimatology. An analogous problem is addressed in the modern ocean by Forget and Wunsch (2007), and Zhao et al. (2018) considered spatial representativeness in choosing how to weight deglacial radiocarbon time

35   series in spatial bin averages. A challenge of any such approach is that the spatial averaging functions (analogous to our $\tau_y$,

but occupying three spatial dimensions) represented by proxy records are often not well known; Van Sebille et al. (2015), for instance, explores how ocean advection determines three-dimensional patterns represented by sediment core observations. Because spatial patterns and time scales of ocean and climate variability are linked, it may ultimately be necessary to consider the full, four-dimensional spatiotemporal aliasing problem.

5    The hope is that these procedures may prove useful for first-order practical uncertainty quantification. A challenge is estimating the signal spectrum $|\hat{r}|^2$, which itself can be affected by aliasing (Kirchner, 2005). One approach is to use spectra from other records that are more highly-resolved or were sampled densely, e.g. from a sediment core at an adjacent site, or a record believed to record similar climate variability. Alternately, measurements of archive properties that can be made cheaply and at high resolution – such as magnetic susceptibility, wet bulk density, and other proxy properties that are routinely made

10    on sediment cores – could prove useful for estimating $|\hat{r}|^2$ if those properties are related linearly to $r(t)$ (Herbert and Mayer, 1991; Wunsch and Gunn, 2003). Another challenge is that time scales that we have shown affect errors are often not published alongside paleoclimate datasets, thus turning systematic errors (where parameters like $\tau_y$ are known) into stochastic errors because a range of possible values must be explored. Publishing all available information about sampling practices, age model construction, and assessments of archive smoothing will greatly aid uncertainty quantification efforts.

15    ## Appendix A: Expressing time-mean temporal representativeness errors in the frequency domain

This appendix describes an analytical approach for estimating temporal representativity errors in the context of estimating time means. These errors have a compact representation in the frequency domain that rationalizes interactions between sampling procedures, time uncertainty, and signal spectra in contributing to errors. Fore more on the theorems and properties of Fourier analysis that are referenced see e.g. Bracewell (1986).

20    ## A1    Derivation

Define a mean value $m(t,\tau)$ of a climate variable $r(t)$ as a function of the duration $\tau$ and the time $t$ on which that duration is centered,

$$m(t,\tau) = \int_{-\infty}^{\infty} \Pi(t',\tau)\, r(t+t')\, dt', \tag{A1}$$

where $\Pi(t,\tau)$ is a normalized "boxcar" function centered on $t=0$ with width $\tau$,

25    $$\Pi(t,\tau) = \begin{cases} 1/\tau & |t| \le \tau/2 \\ 0 & |t| > \tau/2. \end{cases} \tag{A2}$$

The operation in (A1) defines a moving average $m(t,\tau)$ and is known as a convolution, hereafter denoted as a star,

$$m(t,\tau) = \Pi(t,\tau) \star r(t). \tag{A3}$$

Then let the target quantity $x$ be a mean of $r(t)$ over an interval of length $\tau_x$ centered on $t$, and an observation $y$ to be an average over a different duration $\tau_y$ centered on a different time $t + \Delta$,

$$x = m(t, \tau_x) \tag{A4}$$

$$y = m(t + \Delta, \tau_y). \tag{A5}$$

The Fourier transform will be written both using the operator $\mathscr{F}$ and by a hat. Denoting frequency by $\nu$, it is defined as

$$\mathscr{F}(x(t)) \equiv \hat{x}(\nu) = \int\limits_{-\infty}^{\infty} x(t) e^{-2\pi i \nu t} dt.$$

Parseval's theorem states that the integral of a squared quantity in the time domain is equal to the integral of the squared
5   amplitude of the Fourier transform of that quantity, so that after substituting (A5) we can write (2) as

$$\langle \theta^2 \rangle = \frac{1}{\tau_0} \int\limits_{-\infty}^{\infty} (m(t, \tau_x) - m(t + \Delta t, \tau_y))^2 dt \tag{A6}$$

$$= \frac{1}{\tau_0} \int\limits_{0}^{\infty} \left| \mathscr{F} [m(t, \tau_x) - m(t + \Delta t, \tau_y)] \right|^2 d\nu. \tag{A7}$$

By the Fourier shift theorem,

$$\mathscr{F} [m(t + \Delta, \tau_y)] = e^{-2\pi i \nu \Delta} \mathscr{F} [m(t, \tau_y)]. \tag{A8}$$

Then, by the linearity of the Fourier transform,

$$\langle \theta^2 \rangle = \frac{1}{\tau_0} \int\limits_{0}^{\infty} \left| \hat{m}(\nu, \tau_y) - e^{-2\pi i \nu \Delta} \hat{m}(\nu, \tau_x) \right|^2 d\nu. \tag{A9}$$

10   By the convolution theorem, convolution in the time domain is equivalent to multiplication in the frequency domain. Thus, the Fourier transform of a time mean as defined in (A3) is

$$\hat{m}(\nu, \tau) = \mathscr{F} [\Pi(t, \tau) \star r(t)] \tag{A10}$$

$$= \hat{\Pi}(\nu, \tau) \cdot \hat{r}(\nu). \tag{A11}$$

Substituting this relation into (A9) yields

$$\langle \theta^2 \rangle = \frac{1}{\tau_0} \int\limits_{0}^{\infty} \left| \hat{\Pi}(\nu, \tau_x) - e^{-2\pi i \nu \Delta} \cdot \hat{\Pi}(\nu, \tau_y) \right|^2 |\hat{r}(\nu)|^2 d\nu. \tag{A12}$$

Finally, we represent smoothing prior to sampling by defining a new climate signal, $r'(t)$, that has had a running mean applied,

$$r'(t) = \Pi(t, \tau_a) \star r(t). $$

15   Substituting $\hat{r}'(\nu)$ into (A12) and applying the convolution theorem gives

$$\langle \theta^2 \rangle = \frac{1}{\tau_0} \int\limits_{0}^{\infty} \left| \hat{\Pi}(\nu, \tau_x) - e^{-2\pi i \nu \Delta} \cdot \hat{\Pi}(\nu, \tau_a) \cdot \hat{\Pi}(\nu, \tau_y) \right|^2 |\hat{r}(\nu)|^2 d\nu. \tag{A13}$$

Numerical integration of (A13) is used in the text to illustrate dependencies of TR error on sampling parameters.

**A2 Interpretation**

The integrand of (A13) is the product of two components. The second, $|\hat{r}(v)|^2$, is the power spectral density of $r(t)$, which describes the variance contained at frequencies in $r(t)$. The first component is a power transfer function,

$$H(v, \tau_x, \tau_y, \tau_a, \Delta) = \left| \hat{\Pi}(v, \tau_x) - e^{-2\pi i v \Delta} \cdot \hat{\Pi}(v, \tau_a) \cdot \hat{\Pi}(v, \tau_y) \right|^2, \tag{A14}$$

which describes how power at different frequencies in $r(t)$ contributes to $\langle \theta^2 \rangle$. The Fourier transform of the boxcar function is a sinc function,

$$\hat{\Pi}(v, \tau) = \text{sinc}(\tau v) = \frac{\sin(\pi \tau v)}{\pi \tau v}, \tag{A15}$$

which converges towards 1 at frequencies below $1/\tau$ and oscillates with decreasing amplitude about 0 at higher frequencies (Figure 2a).

When $\tau_x$ and $\tau_y$ are adequately separated so that the transfer function has a simple bandpass shape as seen in Figure X, the "cutoff frequencies" $v_{low}^{\dagger}$ and $v_{high}^{\dagger}$ are useful to diagnose how sampling procedures contribute to TR error. These are the frequencies on either side of the band at which the transfer function is equal to 0.5. In the presence of zero time offsets, the cutoff frequencies can be estimated by solving

$$H\left(v_{low}^{\dagger}\right) \approx \left| \text{sinc}^2\left(\tau_x v_{low}^{\dagger}\right) - 1 \right|^2 = \tfrac{1}{2} \tag{A16}$$

$$H\left(v_{high}^{\dagger}\right) \approx \left| \text{sinc}\left(\tau_y v_{high}^{\dagger}\right) \right|^2 = \tfrac{1}{2}. \tag{A17}$$

which yields $v_{low}^{\dagger} = 0.755 \tau_x^{-1}$ and $v_{high}^{\dagger} = 0.443 \tau_y^{-1}$. (In the case where $\tau_x$ is less than $\tau_y$, then $v_{low}^{\dagger} = 0.755 \tau_y^{-1}$ and $v_{high}^{\dagger} = 0.443 \tau_x^{-1}$).

We can expect the presence of archive smoothing might reduce errors originating from high frequencies in $r(t)$, thereby reducing $v_{high}^{\dagger}$ and narrowing the band of aliased frequencies. In the presence of archive smoothing, the expression for $v_{high}^{\dagger}$ becomes

$$H\left(v_{high}^{\dagger}\right) = \left| \text{sinc}\left(\tau_a v_{high}^{\dagger}\right) \text{sinc}\left(\tau_y v_{high}^{\dagger}\right) \right|^2 = \frac{1}{2}. \tag{A18}$$

An approximate solution using a Taylor series representation is

$$v_{high}^{\dagger} \approx \frac{0.443}{\sqrt{\tau_a^2 + \tau_y^2}}, \tag{A19}$$

which illustrates a combined effect from sampling and archive smoothing for determining which frequencies contribute to TR errors. Thus when $\tau_y$ and $\tau_a$ are small relative to $\tau_x$, archive smoothing reduces TR errors, consistent with numerical integrations (comparing Figures 5a and 5b with 5c and 5d).

Using (A19), we can estimate an ideal sampling interval $\tilde{\tau}_y$ in the presence of archive smoothing by minimizing the width of the frequency band that contributes to TR error. Setting $0.443 \tilde{\tau}_x^{-1}$ (i.e., the cutoff frequency in the case where the combined

averaging effect of sampling and smoothing gave the desired averaging interval $\tau_x$) equal to $0.443(\tau_y^2 + \tau_a^2)^{-\frac{1}{2}}$ and solving yields

$$\tilde{\tau}_y = \sqrt{\tau_x^2 - \tau_a^2} \text{ for } \tau_x > \tau_a. \tag{A20}$$

Numerical experiments (see dotted lines in all panels of Figure 5) support the robustness of this approximation for two different signal spectra.

[Figure]

**Figure A1.** Illustration of the frequency dependence of errors in representing an instantaneous measurement of a process $r(t)$ at a time $t$ by another measurement $r(t+\Delta)$. Each line represents a different frequency component of $r(t)$, grey vertical lines represent sampling times, and colored circles represent values of components at those times. At frequencies $\nu = \frac{n}{\Delta}$ for $n = 0, 1, 2, \ldots$, (a), the Fourier components of $x(t)$ will be exactly in phase when sampled at a time lag $\Delta$, so these components do not contribute to the error variance $\left\langle (r(t) - r(t+\Delta))^2 \right\rangle$. By contrast, at frequencies $\nu = \frac{n}{\Delta} + \frac{1}{2\Delta}$ (b), the Fourier components are exactly out of phase, so these components tend to contribute most to the error variance. At intermediate frequencies, contributions lie between the two extremes, leading to a cosine function of error contribution as a function of frequency (Equation A22).

To study the error contribution from a time offset $\Delta$, consider the limit where $\tau_x$, $\tau_y$, and $\tau_a$ approach zero, corresponding to instantaneous observations in time, so that $\langle\theta^2\rangle$ approaches

$$\langle\theta^2\rangle = \frac{1}{\tau_0}\int_0^\infty \left|1 - e^{-2\pi i v\Delta}\right|^2 |\hat{r}(v)|^2\, dv. \tag{A21}$$

Expanding $\left|1 - e^{-2\pi i v\Delta}\right|^2$ and simplifying gives

$$\langle\theta^2\rangle = \frac{1}{\tau_0}\int_0^\infty \left(2 - 2\cos\left(2\pi v\Delta\right)\right)|\hat{r}(v)|^2\, dv \tag{A22}$$

so that the power transfer function is $H = 2 - 2\cos\left(2\pi v\Delta\right)$ and the expected error due to $\Delta$ is a cosinusoidally-weighted
5   function of the signal power spectrum. $H$ takes a minimum value of 0 at frequencies

$$v_{min} = 0, \frac{1}{\Delta}, \frac{2}{\Delta}, \dots \frac{n}{\Delta}$$

for integer values of $n$; at these frequencies, measurements spaced by $\Delta$ in time are in phase and are therefore exactly correlated (Figure A1a). The weights take a maximum value of 4 at frequencies

$$v_{max} = \frac{1}{2\Delta}, \frac{3}{2\Delta}, \frac{5}{2\Delta}, \dots \frac{n}{\Delta} + \frac{1}{2\Delta}$$

where measurements separated by $\Delta$ are always exactly out of phase (Figure A1b). At those frequencies, the underlying signal $r(t)$ is projected twofold onto the error, so that its variance contribution is multiplied fourfold. These variations in frequency
10   contributions modulate effects from smoothing and sampling timescales (Figure 2b).

**Appendix B: Expressing time series temporal representativeness errors in the frequency domain**

This appendix extends the analytical approach for estimating temporal representativity errors from estimating time means to time series. Define the moving average time series associated that would result if all of $r(t)$ were sampled as the $i^{th}$ observation $y^i$ to be

15   $$y^i(t) = \Pi\left(t, \tau_y^i\right) \star \Pi\left(t, \tau_a^i\right) \star r(t) \tag{B1}$$

where we have included a contribution from archive smoothing, so that its Fourier transform is

$$\hat{y}^i(v) = \hat{\Pi}\left(v, \tau_y^i\right) \cdot \hat{\Pi}\left(v, \tau_a^i\right) \cdot \hat{r}(v). \tag{B2}$$

By Shannon's sampling theorem, an accurate discrete representation of $r(t)$ results from sampling all frequencies in $r(t)$ less than or equal to the local Nyquist frequency $v_{Nyq}^i = 1/\left(2\tau_s^i\right)$. As such, the target value $x^i$ for the $i^{th}$ measurement $y^i$ is the value
20   of $r(t)$ sampled at $t^i$ after filtering $r(t)$ to remove high-frequency variability. The Fourier transform of a time series of values of $x^i$ is

$$\hat{x}^i(v) = G\left(v, \tau_s^i\right)\hat{r}(v) \tag{B3}$$

where the "ideal" transfer function $G(v, \tau_s)$ is the Heaviside function

$$G(v, \tau_s) = \begin{cases} 1 & v < 1/(2\tau_s^i) \\ 0 & v \geq 1/(2\tau_s^i) \end{cases} \tag{B4}$$

that is ideal in the sense that it eliminates variability at frequencies greater than $v_{Nyq}^i = 1/(2\tau_s^i)$. Then we define TR error at the $i^{th}$ measurement to be

5   $\theta^i = x^i - y^i.$ (B5)

As in the previous section, we estimate the variance of $\theta^i$ by taking the expected value as if the entire record had been sampled using the local values $\tau_s^i$ and $\tau_y^i$. Then, equivalent to (A13),

$$\langle \theta^{i2} \rangle = \frac{1}{\tau_0} \int_0^\infty \left| G(v, \tau_s^i) - \hat{\Pi}(v, \tau_a^i) \cdot \hat{\Pi}(v, \tau_y^i) \right|^2 |\hat{r}(v)|^2 \, dv. \tag{B6}$$

*Acknowledgements.* Thanks to LuAnne Thompson, Greg Hakim, Lloyd Keigwin, Cristi Proistecescu, Carl Wunsch, and Thomas Laepple

10   for useful conversations. Comments from two anonymous reviewers in a previous round of reviews helped improve the manuscript. Support came from NOAA grant NA14OAR4310176 and an NSF postdoctoral fellowship. Wavelet software was provided by C. Torrence and G. Compo, and is available at URL: http://paos.colorado.edu/research/wavelets/.